# Emotion-Based Literature Book Classification Using Online Reviews

**Elena-Ruxandra Luţan *** and **Costin Bădică ***

Department of Computers and Information Technology, University of Craiova, 200585 Craiova, Romania
* Correspondence: elena.ruxandra.lutan@gmail.com (E.-R.L.); costin.badica@edu.ucv.ro (C.B.)

**Abstract:** Reading is not only a recreational activity; it also shapes the emotional and cognitive competences of the reader. In this paper, we present a method and tools for the analysis of emotions extracted from online reviews of literature books. We implement a scraper to create a new experimental dataset of reviews gathered from Goodreads, a website dedicated to readers that contains a large database of books and readers' reviews. We propose a system which extracts the emotions from the reviews and associates them with the reviewed book. Afterwards, this information can be used to find similarities between the books based on readers' impressions. Lastly, we show the experimental setup, consisting of the user interface developed for the proposed system, together with the experimental results.

**Keywords:** natural language processing; Web scraping; emotion analysis

## 1. Introduction

Nowadays, customer feedback plays a very important role for any producer of goods or provider of services. All companies aim to offer high-quality products that customers enjoy and are more likely to recommend to other peers, as this leads to attracting more and more customers, thus bringing an increased direct benefit for the well-being of the company.

E-marketing and digital branding techniques are highly used to promote products, but many studies have shown that the customers are usually mostly attracted by word-of-mouth [1] when making their purchase decisions. It has been observed that when a customer is satisfied with a company or product, he or she might tell other people about his or her experience. On the other hand, if the customer is dissatisfied, he or she is more likely to share the experience to all his acquaintances [2]. Considering this, companies are advised to regularly check their customers' feedback in order to see why customers are not content with their goods or services and, consequently, to improve in these sections.

In this paper, we focus on a specific category of products, literature books, and we are interested in analyzing the customer feedback related to them—book reviews.

Studies have shown that peers' opinions of a book is one of the top criteria on which people choose their next book to read [3,4]. Online reviews are very handy for readers who are interested in knowing about other peoples' experiences with reading certain books. This is because the reviews appreciate, critique or summarize the book, therefore giving the user the possibility to become familiar with the subject of the book and determine if it will be an enjoyable read [5].

In this context, we consider that it would be both interesting and useful to grasp how different aspects of online reviews, such as the emotions hidden behind the reviewers' words, contribute as general features of the book. Our aim is to identify the emotions triggered by reading books that are captured from online book reviews and use them in order to define an emotion-based categorization of books.

The main contribution of this paper is a new approach to book classification based on modeling and extracting emotions from book reviews. In particular, we provide new

insights on how relevant the emotions are sentiment-wise by comparing them with the ground truth provided by the scaled rating attached to the review by the review author. Our results show that the emotions extracted from the reviews can be considered additional features of the book, and together with other relevant information, such as author and genre, can be used in our future work to generate better book recommendations.

Although our proposed method is not so computationally elaborate as other supervised ML techniques (including, for example, NN approaches), this is actually an advantage, as it offers good performance and low computation cost for accurately retrieving emotions from the reviews. In particular, there are no training and labelling costs as compared with the more complex supervised approaches.

We also propose an experimental system that extracts the emotions present in literature book reviews and assigns them as features of the reviewed book.

The paper is structured as follows. In Section 2, we present related works. Section 3 describes the design of the system, consisting of gathering the reviews, preprocessing the reviews' content, and extracting the emotions. In Section 4, we show the dataset overview and discuss the experimental results. Section 5 presents the conclusions and future work.

## 2. Related Works

Product reviews capture customers' evaluations of products or services. Online reviews are usually presented in a text form next to the product or service description on a website, sometimes accompanied by images and videos taken to support the review [6]. These reviews represent an important source of information for the provider, as well as for other potential customers, because they summarize the reasons for liking or disliking a product or service from the user perspective.

Performing an automated analysis of reviews assumes the availability of datasets that can be used for experiments in order to extract the user sentiments and emotions from the text. Therefore, we checked different datasets of books and reviews on the Internet. Ref. [7] offers a dataset of the 10 thousand most-rated books on the Goodreads website. The number of books and ratings in the dataset is impressive, but it does not help us because it only provides the number of ratings and not the review content. Ref. [8] provides a very good set of datasets for academic use, consisting of books, reviews and user information in JSON format, but it constrains us in choosing a certain book domain on which to perform the analysis.

After a short analysis, we concluded that existing datasets either did not include all the necessary details or the form in which the data were presented did not meet our needs and expectations. In this situation, we concluded that we needed to create our own scraping system to extract the necessary data from a website.

An analysis of how the books are classified into categories on the Goodreads website is provided by Melanie Walsh and Maria Antoniak in [9]. The Goodreads website offers a collaborative tagging system in which the users can group the books into virtual shelves; afterwards, these shelves or tags are used to classify the book into different literature genres. The article focuses on what determines the classification of a book as a "classic". Although limited in scope, this analysis still gave us a good background on the shelving system, the evolution of the website, and certain features of the reviews.

The same authors offer a scraper implementation [10], which collects the data used in the article [9] and stores it in JSON files. We consulted this implementation when developing our scraping system with regard to how we can overcome the problems caused by the windows popping up when loading the Goodreads pages, which hindered the automated scraping process.

In the article [11], M. Colhon et al. presented a method of analyzing sentiments of tourists' reviews. The dataset for experiments was extracted from the AmFostAcolo website, which is a Romanian website where tourists can share their travelling experiences. The authors proposed a method to compute the polarity of the review by counting positive and negative words inside the review. In order to define the positive and negative words,

an English lexicon of words representing emotions was considered and translated into Romanian.

The research of M. Malik and A. Hussain [12] focused on the importance of emotions embedded in online reviews. They defined eight basic emotions for the text: four positive (joy, surprise, anticipation, trust) and four negative (angry, anxiety, sadness and disgust). For mapping the words from reviews with their corresponding emotions, they used a word-emotion association lexicon presented by The National Research Council Canada. For experiments, two different datasets were used, both referring to reviews of products belonging to a mixture of categories. The study highlights how emotions contribute to the helpfulness of online reviews.

An approach of classifying food restaurants using opinion mining was proposed by Y. Kumar et al. in [13]. The polarity of opinions is computed based on the attitude of public audiences or individuals in order to rate the review as positive or negative. The dataset for the experiments was retrieved from Kaggle, and it was split in two parts: 70% of reviews were used for training the data model, and the remaining 30% were used for making predictions. To compute the polarity of the review, the Text Blob library was used. The proposed model counts the words from the positive and negative reviews and assigns a rate to each word. These rates are later used to predict the polarity of a new review belonging to the test dataset.

Different methodologies of sentiment analysis of book reviews were presented by A. Mounika and Dr. S. Saraswathi in [14]. The authors described the stages of the sentiment analyzer which lead to categorizing the reviews into clusters. The clusters of reviews and their authors (users) can be used by a recommender system to provide personalized information.

K.S. Srujan et al. proposed a different approach based on supervised machine learning algorithms in [15] for classifying book reviews extracted from Amazon website. The idea was to map the reviews into numerical vectors based on techniques inspired from information retrieval and then apply machine learning classification algorithms to determine the sentiment score assigned to a review: positive, negative or neutral.

In [16], Valentina Franzoni et al. made a comparison between the emotions extracted from book blurbs and the emotion tags assigned by users through reviews. The dataset was extracted from Zazie, an Italian social network similar to Goodreads, which introduced emotional icon tagging as a new dimension for book descriptions. The idea of the research was to see if the emotions extracted from book blurbs are similar to the tags provided by the users and if an automated classification of books is possible with acceptable accuracy.

The authors R. Ganda and A. Mahmood [17] provided a model that uses pre-trained word vectors for sentence-level classification tasks together with recurrent neural networks. Although recurrent neural networks are efficient at capturing the semantics of the sentences, the computation incurred is a time-consuming task. The authors considered that using word vectors as an extra feature with recurrent neural networks can increase the performance of the system.

Zeng et al. [18] proposed an unsupervised model for sentiment classification that uses pairs of words composed of opinion words and target words. Using dependency parsers and a set of rules, the target-opinion words are extracted from reviews. The goal was to predict an opinion word given a target word.

In [19], David Robinson classified basic emotions depending on three attributes of the emotional experience and personality: emotions motivating a subjective quality, emotions that appear as a result of a certain event, and emotions that motivate a particular kind of behavior. Although the emotions were divided into detailed categories according to the three criteria of the mental experience, we are only interested into the subdivision of "positive emotions" and "negative emotions" presented in the article, which helps us classify our list of emotions into two categories.

The Human-Machine Interaction Network on Emotion (HUMAINE) [20] has proposed an emotion annotation and representation language (EARL) which classifies 48 emotions

into the following categories: *negative and forceful, negative and not in control, negative thoughts, negative and passive, agitation, positive and lively, caring, positive thoughts, quiet positive, and reactive*. However, similarly to [19], we are interested only in the positive-negative classification, regardless of the other subcategories.

Emotions by groups is a concept developed by P. Shaver et al. [21] and also featured by W.G. Parrot [22]. This refers to the fact that starting from six primary emotions (*love, joy, surprise, anger, sadness, fear*), one can define related secondary and tertiary emotions. We use these emotion groups in order to map the secondary or tertiary emotions we have in the emotion file with the positive and negative emotions given by [19,20].

## 3. System Design

Conducting emotion analysis on a text refers to classifying the text based on the emotion carried by the words of the text. Then, this information can be used for different purposes, such as computing the emotions transmitted by a book, classifying the books into certain categories based on the emotions, or making recommendations of books that transmit similar sentiments. Our aim is to build a system for literature book classification based on the emotions that are present in online reviews of the books.

Figure 1 shows the system workflow containing the two main tasks performed by the system:

1. Data extraction from the Goodreads website and storage into CSV files;
2. Extraction of the emotions present in the reviews.

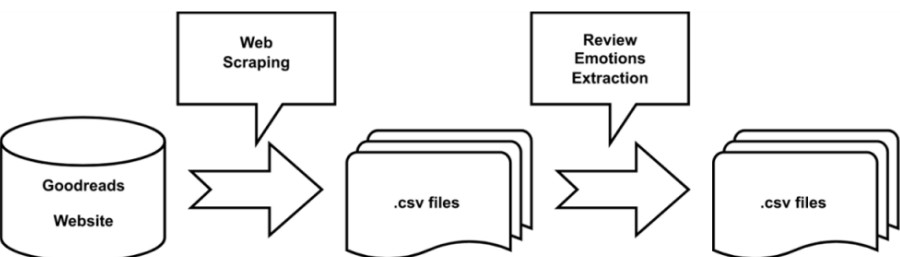

**Figure 1.** System workflow.

We propose the use of datasets extracted from the Goodreads website, which has a very large database. It is considered to be the world's largest website for readers and book recommendations with more than 3.5 billion books, 80 million comments and 125 million users [23]. It contains almost every existing book, and, depending on the popularity of the book, there are a great number of reviews to choose from.

After our initially visual and then more detailed manual inspection of the Goodreads Web pages, we noticed that their HTML content can be quite easy collected. The only disadvantage identified is that Goodreads does not provide much information about the users (book readers) that can be retrieved. This is caused by either the fact that either the user account is set to private or the user did not publish much information about him or herself. Because of this, we focused our analysis only on the contents of reviews and not on the users' information.

Three main entities that are relevant for our goal were identified on Goodreads Web pages: books, reviews, and users. For each entity, multiple attributes were extracted using Web page scraping. These attributes are illustrated in Figure 2.

Based on these entities, three different files are created using Web scraping, i.e., for each entity and its attributes, a separate file was created.

The goal of the first task was to collect the experimental dataset. We created a Python application to scrape the content of Goodreads website using Beautiful Soup [24]. We used a separate text file to specify which books have to be collected. For this, we wrote a book title on each line. The application goes through each line of the file and conducts a search on the Goodreads website using the available words in the book description. After accessing

the page with the book specifications, the parser analyses the HTML content, extracts the fields of interest for the book description and creates a dictionary entry, which is stored inside the database of books.

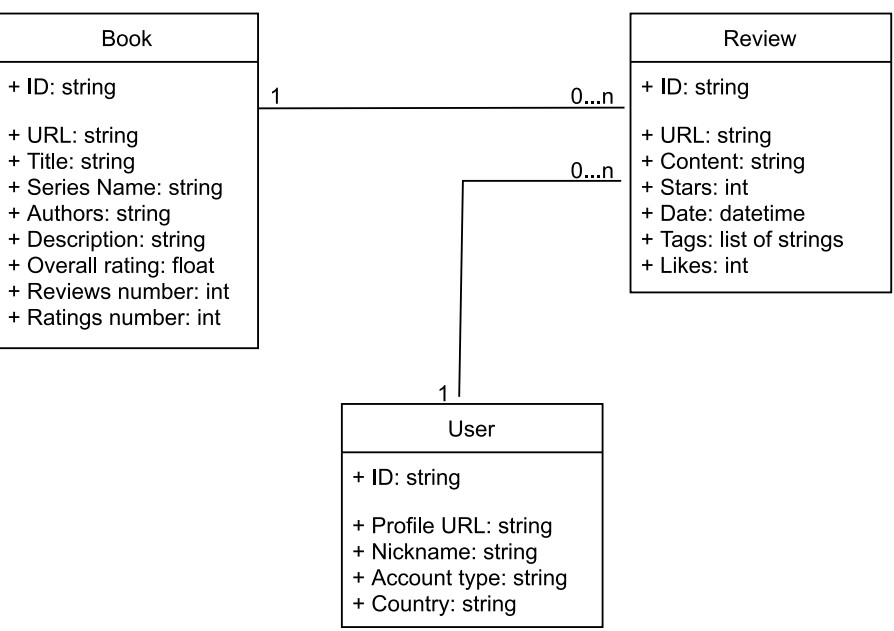

**Figure 2.** Entity and their relationships on Goodreads Web pages.

During its next step, the parser checks the comments section of the page in order to collect the reviews as well as other users' data. The fields that are retrieved for the two entities are described in Figure 2. By default, the Goodreads reviews are sorted by popularity, which is assessed by the reactions of other users to the respective reviews. On the website, a user has the possibility to react to a review by using a "like" button or by adding comments. We think it is an advantage for us to collect the reviews sorted by popularity, as we can assume that the most relevant reviews for analysis are the ones with the most reactions.

After the data are collected, we can start the sentiment and emotions analysis of the reviews. This refers to being able to classify the reviews as positive, negative or neutral and extracting the emotions from the text. The model uses a dimension of 35 emotions and a list of words associated with these emotions.

In order to perform a sentiment and emotion analysis of a text, it is necessary to prepare it by converting all letters into lowercase, removing punctuation and stop words. During the text preprocessing, we removed additional parts from the text, such as: duplicated letters to exaggerate the words (e.g., "ahhhhh", "I loooooove"), insertions of elements to create an atmosphere when reading the comment (e.g., *finally*, *spoiler alert*), and insertions of quotes from the book, URLs or emails. Algorithm 1 describes the complete process of preprocessing the review content for sentiment and emotion analysis.

---

**Algorithm 1** Algorithm for preprocessing (cleaning) the review content

---

1: Convert letters to lowercase
2: Remove punctuation by replacing it with spaces
3: Remove duplicate letters used for exaggeration
4: Remove notes written between ** and ()
5: Remove quotes from the book
6: Remove URLs and emails
7: Remove new lines, tabs, multiple space characters
8: Remove stop words

---

Using the clean text, we compute the polarity of the review. The polarity is a real number in the range $[-1, 1]$, where 1 means positive and $-1$ means negative. By using the polarity, we can classify the comment as being positive (polarity > 0), negative (polarity < 0) or neutral (polarity = 0).

The polarity is computed using TextBlob, which is a Python library that offers a simple API to perform shallow NLP tasks [25]. The library uses a dictionary of English adjectives such that each adjective has a polarity value assigned according to its emotion. The overall polarity of the review is computed based on the polarities of the adjectives present in the review. Afterwards, the review polarity, together with the review classification (positive, negative or neutral) are added as new columns inside the review dataset.

The NLP Emotion Algorithm used in the project is inspired by [26]. It consists of cross-checking all the words of the review (clean text) with the words present inside an emotions file called *emotions.txt*. The emotions file contains a series of 517 adjectives from the English language. Each adjective is assigned an emotion. In total, there are 35 emotions considered: *'cheated', 'singled out', 'loved', 'attracted', 'sad', 'fearful', 'happy', 'angry', 'bored', 'esteemed', 'lustful', 'attached', 'independent', 'embarrassed', 'powerless', 'surprise', 'fearless', 'safe', 'adequate', 'belittled', 'hated', 'codependent', 'average', 'apathetic', 'obsessed', 'entitled', 'alone', 'focused', 'demoralized', 'derailed', 'anxious', 'ecstatic', 'free', 'lost', 'burdened'*.

In the implementation proposed by [26], each adjective from the *emotions.txt* file is checked for existence inside the review in order to compute the emotions list corresponding to the review. If the word is present in the review, the associated emotion is added into the emotions list. After analysis, for each review, we obtain a list of emotions in form of a counter object. This counter is useful because it provides us an image about the weight of a certain emotion inside the review.

Although the implementation [26] was able to extract emotions from the reviews, we realized that it provides only a light overview regarding the emotions present in the review.

The main disadvantage we observed is regarding repetitive words expressing emotions inside the review. In [26], the approach was to check if each adjective from *emotions.txt* is present inside the review, returning true or false output. This means that in case an adjective is used multiple times inside the review, the emotion expressed will only be counted once based on its first occurrence of the adjective. When conducting an emotion analysis, we consider this does not provide a good overview, because if a certain adjective occurs multiple times, the intensity of that emotion shall be taken into consideration in the analysis by increasing the weight of the respective emotion.

As a result of this observation, we decided to change the method of matching the words from the reviews with the adjectives from *emotions.txt* file. In our implementation, we split the review into words, check each of the words for presence inside *emotions.txt* and count the corresponding emotion if a match is found. In this way, each occurrence of emotion-relevant words inside the review will be taken into consideration when creating the emotion list of the review.

At this stage, we identified another aspect which can be improved regarding the counting of emotions. Because the *emotions.txt* file contains only adjectives, we are constrained by the occurrence of certain adjectives inside the review in order to determine the emotions. After an analysis, we observed that we can perform an approximate match regarding the words from the review and the adjectives from the *emotions.txt* file and therefore extract more detailed emotions specification.

As an example, before the approximate match update, the comment "I love this book. I appreciate the vision of the author." does not detect any emotion, because the *emotions.txt* file contains adjectives, and no exact match with the existing words can be done. On the other hand, after our improvement, the verb "love" was matched with the word "loved", and the verb "appreciate" with the word "appreciated", therefore extracting the associated emotions "esteemed attached loved".

Algorithm 2 shows the procedure used for extracting the emotions present inside a review.

With these changes, the time needed to analyze the comments was greatly increased, but we consider this a good compromise for obtaining a more detailed analysis of emotions of the comments.

---

**Algorithm 2** Algorithm for extracting the emotions present in a review.

---

1: Clean the review content using Algorithm 1
2: Create an empty emotions list
3: **for** each *word* in review content **do**
4: 　　**if** *word* matches adjective in *emotions.txt* file **then**
5: 　　　　Extract the emotion associated with the *word*
6: 　　　　Add the emotion inside the emotions list
7: 　　**end if**
8: **end for**
9: Create a counter based on emotions list to reflect the weight of each emotion

---

The final step of the emotions analysis is to associate all the emotions found inside the reviews with the books. This refers to going through all the reviews for a book and creating an emotions list for the book by counting all the reviews' emotions and attaching them to the book.

## 4. Experiments and Discussions

### 4.1. Dataset Overview

For our project, we considered that it is important to execute the book selection in such way as to create a mini-universe to which we can apply the sentiment analysis algorithm in order to obtain a rich classification of the books based on their computed emotion-focused similarities.

The most important aspect for us when choosing the data for analysis is to have a large enough number of reviews. Therefore, we decided to select the books based on the top popular books available on the Internet. We used as inspiration the list of books provided in the article "100 books everyone should read before they die (ranked!)"[27].

In Table 1, we present a few quantitative figures regarding our dataset.

**Table 1.** Statistics of dataset collected from Goodreads website.

| Entity | Number |
|---|---|
| Books | 78 |
| Reviews | 6566 |
| Users | 2661 |

One of the attributes collected for each book is "review numbers", which represents the number of reviews available on Goodreads website for the respective book. In Figure 3, we can observe that 86% of the books contain more than ten thousand reviews.

Instead of a clear book categorization into genres, the Goodreads website uses the approach of shelves and tags to classify and organize the books. This shelves system is a collaborative tagging system where the users can give different tags for the same content (book) in order to categorize it. For each book, we considered the first 10 tags to determine the genre.

We created a word cloud to visualize the overall genre distribution of the books we have inside the dataset. This can be seen in Figure 4. We can see that most of the books belong to the same categories: classics, fantasy, fiction etc.

Note that, because of the collaborative tagging system, some of the genres are repeated, with small changes in naming, such as "Academic Read"—"Academic School", "Literature 19th"—"19th Century".

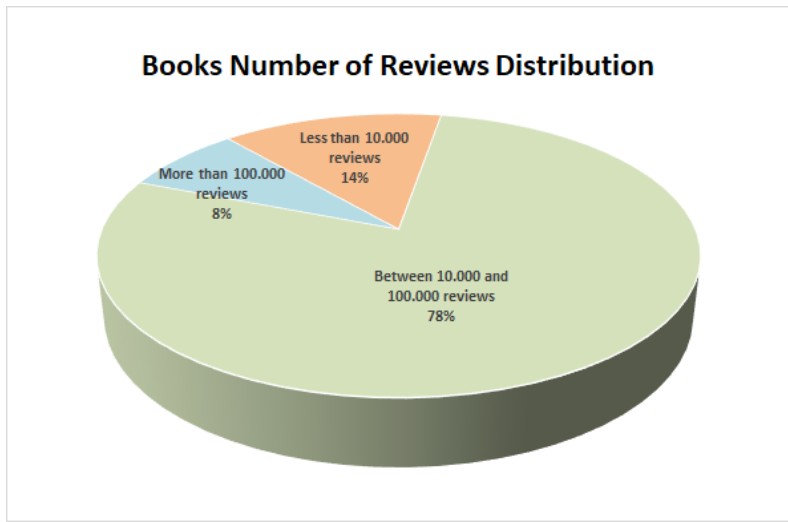

**Figure 3.** Books classification considering the number of reviews available on Goodreads website.

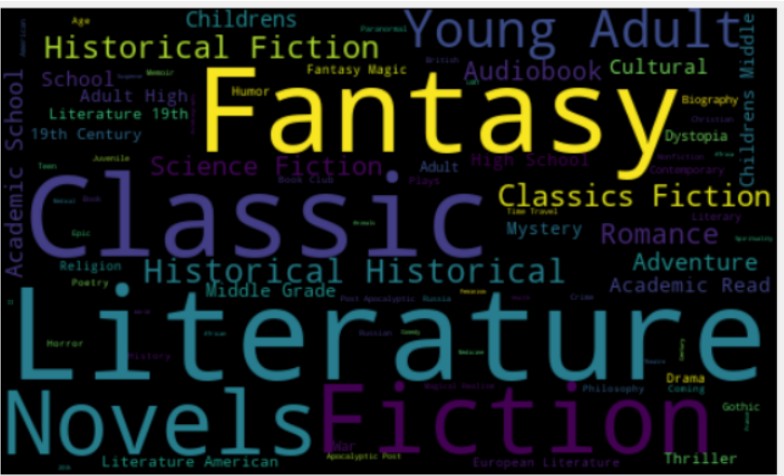

**Figure 4.** Genres of the collected books.

*4.2. Experimental Results*

We assumed that when doing the search, the first book that occurs in the search outcome will be the one we were looking for, since we are using well-known books, which should be at the top of the search. In practice, this depends on the query string used to perform the search. More specifically, we assumed that by using the combination of "book title" and "book author" for the search query, we would obtain the desired book. However, we noticed that rather often, other related books (especially literature guides to the books) were included in the list search results before the sought-after book.

Regarding the review dataset, we noticed that some of the reviews contained information written in languages other than English. Since we use the English language for the text analyzer, we had to remove comments and eventually parts of comments which were not written in English.

Another issue observed was that some of the comments, usually smaller ones, contained information which was not relevant for sentiment analysis or even had nothing to do with the book. For example, some of the reviews contained a short description of the pros or cons of the book, while others contained only links to review videos on other platforms.

We observe that the positive reviews are dominant in the dataset, with a percentage of 86%, followed by the negative reviews, representing 11% of the dataset; only 3% of the dataset consisted of neutral reviews (Figure 5). We consider that this distribution is also influenced by the way we collected the reviews. In the previous sections, we described that

when collecting the reviews, we took the reviews ordered according to the default order present on Goodreads website; therefore, we expect the majority of reviews to be positive ones to influence the visiting user to read the respective book.

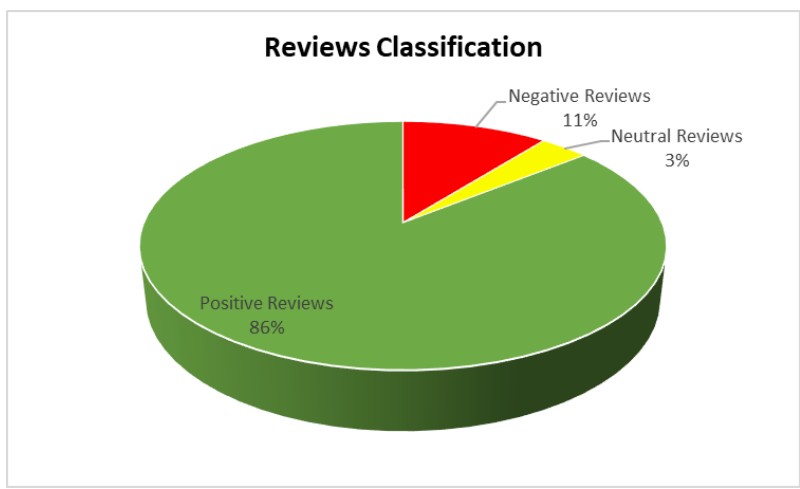

**Figure 5.** Classification of the reviews using polarity value.

In addition, we made an analysis consisting of the polarity classification combined with the number of stars given to the reviews, which can be seen in Figure 6. This is relevant in order to see the false positive or false negative classifications.

We can observe that we have a lot of reviews that received 1 or 2 stars, which were classified as "positive", although from the star number, we can consider that the user actually did not greatly enjoy the book. These are considered false-positive results. The number of false-negative results is lower, as we can see that a large part of the reviews with 3, 4 or 5 stars were classified as "positive", as expected.

Moreover, we also have reviews for which no star-scaled classification was done by the reviewer (column "0"). It can be seen that majority of these comments are positive ones, but we cannot consider them as good inputs for our sentiment analysis system.

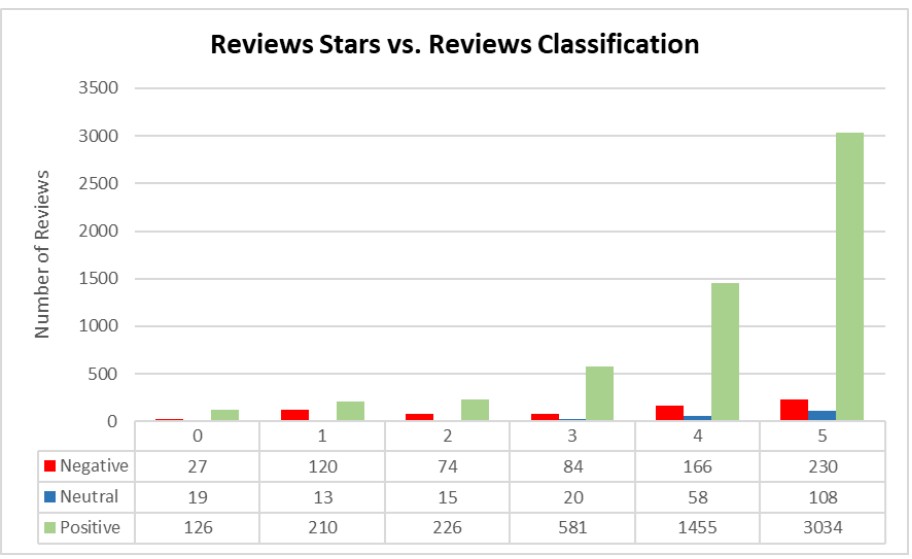

**Figure 6.** Classification of the reviews using review stars and polarity.

Once the sentiments are extracted from the reviews, the sentiments are stored as an additional column inside the review dataset so that it can be easily accessed for future purposes. We decided to store the sentiments as counter objects because in this way we can

see weight of each sentiment inside the review. An extract of the emotions column can be seen in Figure 7.

```
Emotions
[(' happy', 7), (' attached', 4), (' fearful', 3), (' sad', 2), (' loved', 2), (' cheated', 2), (' fearless', 1), (' ecstatic', 1), (' attracted', 1), (' codependent', 1), (' lustful', 1), (' hated', 1), (' angry', 1)]
[(' cheated', 7), (' attached', 5), (' happy', 4), (' fearful', 3), (' lustful', 3), (' fearless', 2), (' powerless', 2), (' sad', 2), (' apathetic', 2), (' entitled', 2), (' loved', 2), (' demoralized', 1), (' ecstatic', 1), (' hated', 1), (' singled out', 1), (' focused', 1)]
[]
[(' cheated', 3), (' sad', 2), (' lustful', 2), (' happy', 2), (' loved', 1), (' powerless', 1), (' obsessed', 1), (' entitled', 1), (' hated', 1), (' apathetic', 1), (' singled out', 1)]
[(' bored', 1), (' powerless', 1), (' attached', 1)]
[(' fearless', 18), (' fearful', 4), (' angry', 4), (' singled out', 3), (' independent', 3), (' cheated', 3), (' focused', 2), (' attracted', 2), (' attached', 2), (' happy', 2), (' entitled', 2), (' apathetic', 1), (' alone', 1), (' burdened', 1), (' esteemed', 1), (' pow
[(' entitled', 14), (' happy', 13), (' sad', 11), (' cheated', 7), (' powerless', 7), (' fearful', 7), (' apathetic', 7), (' angry', 6), (' attracted', 5), (' hated', 3), (' loved', 3), (' singled out', 2), (' fearless', 2), (' belittled', 2), (' focused', 2), (' esteemed', 1), (
[(' attracted', 8), (' sad', 8), (' angry', 5), (' happy', 5), (' attached', 5), (' powerless', 4), (' fearful', 4), (' hated', 4), (' average', 3), (' entitled', 3), (' singled out', 3), (' ecstatic', 2), (' alone', 2), (' codependent', 2), (' burdened', 2), (' fearless', 1),
[(' attached', 4), (' loved', 4), (' hated', 2), (' fearless', 1), (' sad', 1), (' apathetic', 1), (' happy', 1)]
[(' powerless', 8), (' cheated', 7), (' entitled', 4), (' attracted', 3), (' happy', 3), (' fearful', 3), (' sad', 2), (' free', 2), (' attached', 1), (' lost', 1), (' esteemed', 1), (' surprise', 1)]
[(' happy', 19), (' fearful', 14), (' sad', 11), (' powerless', 9), (' burdened', 9), (' angry', 9), (' attached', 8), (' anxious', 8), (' hated', 6), (' loved', 6), (' lustful', 4), (' apathetic', 4), (' safe', 4), (' entitled', 4), (' belittled', 1), (' attracted', 1), (' adequa
[(' entitled', 4), (' happy', 4), (' attached', 3), (' fearless', 2), (' alone', 2), (' apathetic', 2), (' powerless', 2), (' cheated', 2), (' demoralized', 1), (' lost', 1), (' attracted', 1), (' codependent', 1), (' esteemed', 1), (' singled out', 1), (' loved', 1)]
[(' happy', 5), (' fearful', 3), (' attracted', 2), (' sad', 2), (' bored', 2), (' apathetic', 2), (' attached', 2), (' entitled', 2), (' loved', 2), (' hated', 1), (' esteemed', 1), (' anxious', 1)]
[(' attached', 6), (' happy', 2), (' loved', 2), (' sad', 1), (' hated', 1), (' free', 1), (' surprise', 1), (' lustful', 1)]
[(' attached', 5), (' loved', 4), (' happy', 2), (' attracted', 1)]
[(' happy', 3), (' cheated', 3), (' powerless', 1), (' attracted', 1), (' free', 1), (' sad', 1), (' apathetic', 1), (' hated', 1), (' attached', 1), (' angry', 1), (' lustful', 1), (' alone', 1), (' singled out', 1), (' loved', 1), (' anxious', 1)]
[(' cheated', 4), (' singled out', 2), (' powerless', 1), (' lost', 1)]
[(' happy', 2)]
```

**Figure 7.** Snapshot of the "emotions" column from the review dataset.

The book emotions were computed by adding the emotions present in all the reviews for the book. The counter of emotions for the book was added as an additional column called "emotions" to the books dataset, similarly to the one for reviews presented in Figure 7.

The last step in our analysis consisted of evaluating the emotions algorithm. For this, we divided the emotions into three categories (positive, negative and neutral) by considering a series of emotion classification studies ([19–22]), as follows: positive emotions—'*loved*', '*attracted*', '*happy*', '*lustful*', '*fearless*', '*ecstatic*', '*esteemed*', '*safe*', '*adequate*', '*focused*', '*entitled*', '*independent*', '*free*', '*attached*'; negative emotions—'*sad*', '*fearful*', '*angry*', '*bored*', '*embarrassed*', '*powerless*', '*surprise*', '*hated*', '*alone*', '*anxious*', '*cheated*', '*singled out*', '*belittled*', '*lost*', '*burdened*', '*alone*', '*demoralized*', '*apathetic*', '*obsessed*', '*derailed*', '*codependent*'; neutral emotions—'*average*', '*free*'.

Using the Eeotions column in the reviews dataset (Figure 7), for each review, we counted the total of positive, negative and neutral emotions whiel also taking into consideration their weight. In order to determine the correctness of our algorithm, we compared the emotion-based classification with the polarity-based classification. This is illustrated in Figure 8.

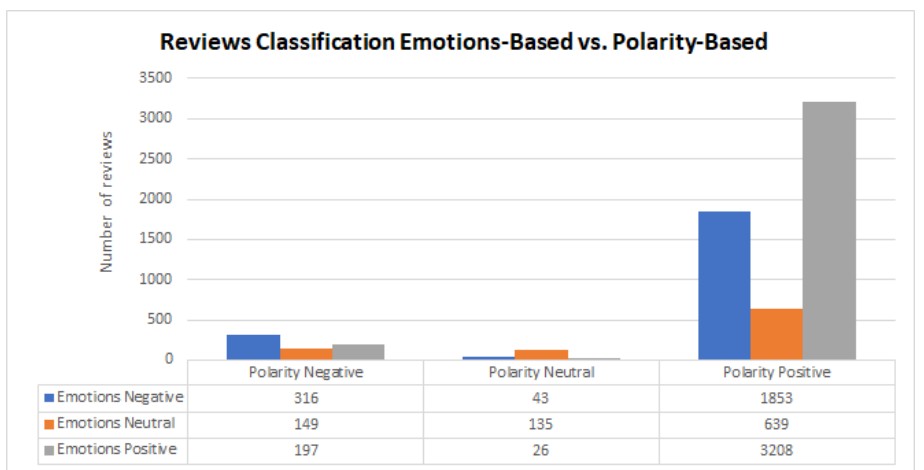

**Figure 8.** Review classes considering emotion-based classification and polarity-based classification.

In Figure 8, we can observe that only 47,73% of the reviews classified as "negative" based on polarity are also classified as "negative" based on emotions. The remaining 52,27% seem to be classified as false positives if we consider that the polarity-based classification was conducted correctly. At a first glance, this would suggest our algorithm does not work correctly, but if we analyze their emotion-based classification with the review stars, we can observe that the emotion-based classification is actually correctly carried out (Figure 9).

The review stars are a valuable input because they are directly given by the user. In Figure 9, we can see that for a considerable number of "negative" reviews (polarity-based), the users have provided 4 or 5 stars, which suggests that the user has liked the book, so he or she actually provided a "positive" review.

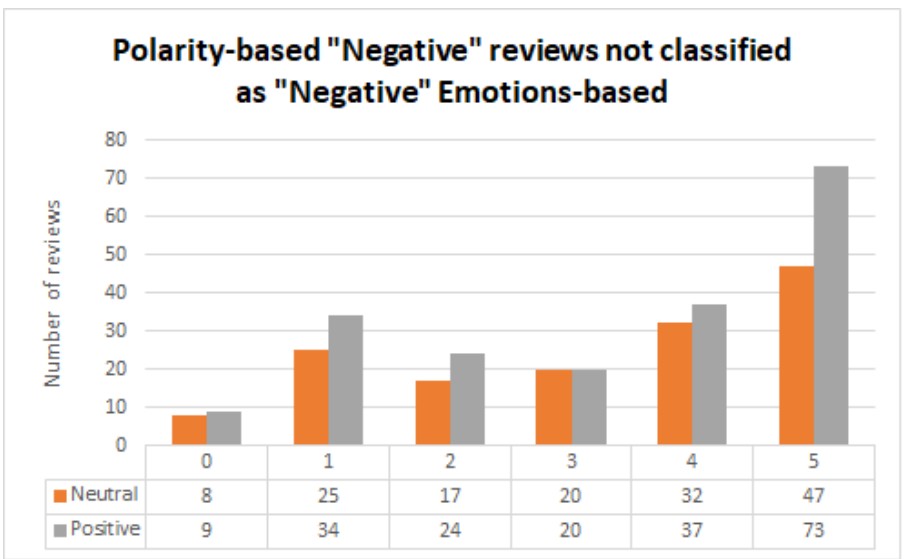

**Figure 9.** Review classes considering emotion-based classification and polarity-based classification.

Let us discuss some examples considering the two types of review classification, which are provided in Table 2. In the case of review 1917, we consider that the classification "neutral" reflects the 3-star rating provided by the user better than the "positive" classification assigned considering the polarity value, while for review 1915, the polarity-based classification better reflects the 4-star rating provided by the user. In review 1916, the user has given a 5-star rating to the book, so the expectation is that we have a positive review, which is the case for both the polarity-based classification and the emotion-based classification. Review 1918 is a clear example of a false-positive classification, because although the polarity-based classification and emotion-based classification are "positive", the used only provided a 2-star rating, which is unexpected.

**Table 2.** Extract of Reviews dataset.

| Review Index | Review Stars | Polarity Classification | Emotions Classification |
|---|---|---|---|
| 1915 | 4 | Positive | Neutral |
| 1916 | 5 | Positive | Positive |
| 1917 | 3 | Positive | Neutral |
| 1918 | 2 | Positive | Positive |

## 5. Conclusions and Future Work

In this paper, we presented a model of extracting the sentiment and emotions of literature books from online reviews. In the first stage, we designed the entities with the fields of interest and created a scraper to collect the dataset from the Goodreads website. The second stage consisted of retrieving the sentiments and emotions present in the online reviews. The sentiments were extracted using the TextBlob Python library. For the extraction of emotions, we used an implementation from Github as a starting point, which we improved in order to fulfill our needs. We re-worked the method of extracting the sentiments to take into consideration verbs or adjectives and arranging the extracted emotions such that they can be easily accessed for different use cases.

As future work, we aim to improve the extraction of emotions from the reviews by considering a larger emotions file or a library for emotions. Another future development

refers to the usage of the emotion-based book classification with a particular scope, such as a recommender system, for providing literature recommendations to users or finding similarities between users who felt the same emotions when reading a certain book.

**Author Contributions:** Conceptualization, E.-R.L. and C.B.; methodology, E.-R.L. and C.B.; software, E.-R.L.; validation, E.-R.L. and C.B.; formal analysis, E.-R.L.; investigation, E.-R.L.; resources, E.-R.L.; data curation, E.-R.L.; writing—original draft preparation, E.-R.L.; writing—review and editing, E.-R.L. and C.B. All authors have read and agreed to the published version of the manuscript.

**Funding:** This research received no external funding.

**Institutional Review Board Statement:** Not applicable.

**Informed Consent Statement:** Not applicable.

**Data Availability Statement:** The source code and datasets are available at: https://drive.google.com/drive/folders/1aGTeWZRd4QK1t48u_0WAazu30Tla-jEc?usp=sharing (accessed on 18 September 2022).

**Conflicts of Interest:** The authors declare no conflict of interest.

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
