# Peer review of "Emotion-Based Literature Book Classification Using Online Reviews"

_electronics, doi:10.3390/electronics11203412_

Round 1

Reviewer 1 Report

Emotion-based Literature Books Classification Using Online Reviews

The authors implemented a scraper to create a new experimental dataset of reviews gathered from Goodreads – a website dedicated to readers that contains a large database of books and readers’ reviews. They proposed a system that extracts the emotions from the reviews and associates them with the reviewed book. Afterward, this information can be used to find similarities between the books based on readers’ impressions.

My comments are given below - 

(1) What I observed is that the authors failed to place their work with respect to the literature. 

(2) It seems to me that this is an implementation of a system, probably an empirical study. If it is the case, the authors should write on that basis. 

(3) How this work can make an improvement with respect to literature? The authors should focus on the introduction to package the whole work nicely and should bring it upfront to get the reader's attention. 

Author Response

(1) What I observed is that the authors failed to place their work with respect to the literature. 

Answer: We revised the Introduction to better reflect the work, its background and purpose. We have added 6 new references to improve the state-of-the-art.

(2) It seems to me that this is an implementation of a system, probably an empirical study. If it is the case, the authors should write on that basis. 

Answer: Indeed, in this research we created a system which performs two kind of tasks: the collection of the dataset from online sources and the processing of the data regarding extraction of sentiments present in the reviews. Our work proposes a new approach for books classification based on modeling and extracting emotions from book reviews. We improved the Introduction section with more details why we consider such system is both interesting and useful.

(3) How this work can make an improvement with respect to literature? The authors should focus on the introduction to package the whole work nicely and should bring it upfront to get the reader's attention. 

Answer: The contribution to literature refers to the fact that although the model uses a simplistic algorithm to extract the emotions from the reviews, the sentiment given by the emotions and the scaled rating of the review are similar, which proves that the emotions outputs of the model can be considered as features of the books and used when comparing a set of books in order to find similarities between them. We improved the Introduction section by describing the main contribution.

Reviewer 2 Report

The introduction should be expanded with more background and some related works.

The overall contribution and innovation are limited. Only using tools to extract sentiment and emotion scores is not enough. Furthermore, the findings are not very valuable for the book reviews recommended field, which does not generate some interesting results and future directions.

The sentiment and emotion extraction methods are common. The authors only use existing word-based basic tools to extract them. The authors should try NN-based models or design novel approaches to book reviews

Author Response

The introduction should be expanded with more background and some related works.

Answer: The introduction was revised and more background was added. In the Introduction section we added description regarding the context and the need of the proposed system. Also, 6 new references were added in the paper, in the Introduction and Related Works chapters.

The overall contribution and innovation are limited. Only using tools to extract sentiment and emotion scores is not enough. Furthermore, the findings are not very valuable for the book reviews recommended field, which does not generate some interesting results and future directions.

The sentiment and emotion extraction methods are common. The authors only use existing word-based basic tools to extract them. The authors should try NN-based models or design novel approaches to book reviews

Answer: Our work proposes a new approach for books classification based on modeling and extracting emotions from book reviews. In particular we provide new insights on how relevant the emotions are sentiment-wise by comparing them with the ground-truth provided by the scaled rating attached to the review by the review author. Our results show that the emotions extracted from the reviews can be considered as additional features of the book, and together with other relevant information such as author and genre, can be used in our future work to generate better book recommendations.

Although the used methods are not so elaborated as other supervised ML techniques (including for example NN approaches), this is actually an advantage, as they offer good performance and low computation costs for accurately retrieving the emotions from the reviews. In particular, there are no training and labelling costs, as compared with the more complex supervised approaches.

We have added in the Introduction section the main contribution and the advantages of our approach compared to other methods from literature.

Round 2

Reviewer 1 Report

I think that the authors have tried to make improvements based on the comments. 

Author Response

Thank you very much for the review!

We have made minor changes to the text with regard to English spell check and modified all diagrams to be in vector format, for better visualization.

Reviewer 2 Report

Reasons for replying are easier to accept.

The figures should be uniform and clear (vector version)

Author Response

(The authors gave the same response as above.)
